# Federated Learning on Patient Data for Privacy-Protecting Polycystic Ovary Syndrome Treatment

**Lucía Morris**[*]
Stanford University
Stanford, USA 94305
luciam@stanford.edu

**Tori Qiu**[⋆]
Stanford University
Stanford, USA 94305
toriqiu@stanford.edu

**Nikhil Raghuraman**[⋆]
Stanford University
Stanford, USA 94305
nikhilvr@stanford.edu

## Abstract

The field of women's endocrinology has trailed behind data-driven medical solutions, largely due to concerns over the privacy of patient data. Valuable datapoints about hormone levels or menstrual cycling could expose patients who suffer from comorbidities or terminate a pregnancy, violating their privacy. We explore the application of Federated Learning (FL) to predict the optimal drug for patients with polycystic ovary syndrome (PCOS). PCOS is a serious hormonal disorder impacting millions of women worldwide, yet it's poorly understood and its research is stunted by a lack of patient data. We demonstrate that a variety of FL approaches succeed on a synthetic PCOS patient dataset. Our proposed FL models are a tool to access massive quantities of diverse data and identify the most effective treatment option while providing PCOS patients with privacy guarantees. Our code is open-sourced at `https://github.com/toriqiu/fl-pcos`.

## 1 Introduction

### 1.1 Privacy Concerns with PCOS Patient Data

Biologic hyperandrogenism is a key criterion for polycystic ovary syndrome (PCOS) diagnosis and is determined by blood work. These blood panels measure various hormones associated with PCOS and provide a snapshot of the patient's overall reproductive health. Blood panel results are sensitive, as they can be used to extract information about pregnancies, abortions, and miscarriages. For example, using the hormone levels revealed by blood panel data, it's possible to access a patient's pregnancy history by detecting menstrual irregularities and using luteinizing hormone (LH) and follicle-stimulating hormone (FSH) to diagnose pregnancy [1], [2]. These hormone levels are especially distinct during early pregnancy [3]. A patient file containing reports of mood disorders or comorbidities, if extracted, could also impact the patient's job prospects and insurance rates [4]. Considering the intimate information PCOS data can reveal about a patient's medical history, privacy is essential when handling it. In situations where medical providers are no longer the sole repositories of patient data, it is not difficult for a third-party server to obtain medical records [5].

### 1.2 Federated Learning Meets the Challenges of PCOS Research

The field of gynecology research largely relies on primary data collection methods and randomized controlled trials (RCTs) for studying diagnosis and treatment. Both these modes of inquiry suffer from a lack of large-scale data. RCTs conducted in OB/GYN studies typically enroll an average study population of 200 female patients [6], which is miniscule compared to the millions of data points

---

[*]All authors contributed equally.

available in aggregated databases, even after filtering for a study's inclusion criteria. However, RCTs are particularly informative because randomizing exposure to treatment eliminates selection bias and rules out confounding variables, establishing causality of a treatment. Despite the advantages of randomization, RCT findings are not generalizable. Their small sample sizes preclude analysis of correlations between hormonal levels and treatment, and they lack guarantees on non-IID data. HIPAA Privacy Rule prohibits U.S. healthcare providers and insurance companies from using or disclosing protected health information without explicit patient consent [7]. Voluntary disclosures are unlikely as widespread anxieties of data security and privacy persist within the patient population [8], and even anonymized data can be re-identifiable via triangulation with other data sets [9], [10]. Not only does the need for patient consent severely limit data availability, it is only practical for cases in which consent granted by the patient is unconditional, since recalling data from those who obtained it is practically unenforceable. These factors complicate the study of PCOS treatment. Ideally, study of PCOS treatment can be conducted at scale with minimal data retention to protect the privacy of subjects.

We propose federated learning (FL) as a method for improving PCOS treatment outcomes. In particular, we develop models to learn optimal oral contraceptive pill (OCP) treatments based on blood panel data and historical success of treatment for similar hormone profiles. We argue that FL is a promising solution to improve the PCOS treatment process given the privacy concerns around patient blood panel data and the lack of comprehensive datasets relevant to PCOS [11]. Isolated data from smaller studies can introduce sample bias in which demographics (e.g. ethnicity, age, etc.) skew the predictions, affecting the accuracy of treatment prediction for certain sites. FL allows access to thousands of valuable data points across a wide geographic area without the need for a centralized database or sharing of data between institutions. Using FL to determine personalized medical treatment bypasses the high risks of direct intervention on patients and the expenses of conducting clinical trials [12]. With FL, the advantages of large datasets can be harnessed for PCOS research without the security threats historically associated with learning at this scale.

### 1.3 Contributions of Our Approach

Current ML research on PCOS focuses on its diagnosis [13], [14],[15], with much of this research employing the same publicly available dataset of 541 points from Kerala, India [16] [17]. Our key contributions are as follows:

- We devise the first application of ML to PCOS drug recommendation, focusing on treatment rather than diagnosis. We learn correlations between seven hormonal metrics associated with PCOS and their compatibility with five common OCPs. Using these correlations, our models predict the most effective intervention.

- We generate synthetic data to reflect the client pool of a deployed FL model, significantly expanding the dataset size from existing studies. The larger dataset helps avoid overfitting to a small group and captures the complete demographics of patients who are impacted by predictive care.

- We demonstrate a variety of FL approaches achieve excellent performance on the task of drug recommendation. Our models are evaluated on IID data, non-IID data, clients with identical dataset sizes, and clients with differing dataset sizes.

## 2 Background and Related Work

### 2.1 PCOS Lacks a Clear Treatment Process

PCOS is the most common hormonal disorder in women of reproductive age, impacting an estimated 10% of women worldwide. PCOS is named for the painful, fluid-filled cysts that form on the ovaries, which can damage organ function and in some cases require surgical removal. Other common symptoms of PCOS include irregular menstruation, infertility, hirsutism, acne, mood disorders, weight gain, and an increased risk for Type II diabetes and heart disease [18].

The precise cause of PCOS is unknown and no cure exists. Using OCPs to regulate hormone levels is the primary method of symptom management among patients [19]. OCPs suppress secretion of luteinizing hormone (LH) and increase sex hormone-binding globulin levels to decrease androgen

and testosterone levels, correlating with fewer PCOS symptoms [20]. The American Society of Reproductive Medicine recommends OCPs as the primary treatment option but provides no guidelines as to which OCPs are effective for various presentations of the disease [21]. There are over 200 OCPs on the market, and the birth control industry has an estimated value of over $13 billion [22]. OCPs containing progestins with lower androgenic activity are most effective in treating PCOS symptoms [19]; examples include Apri, Cyclen, Tri-cyclen, Yaz, and Diane-35.

Typically, a PCOS patient will use a trial and error approach to test various OCPs for a 3-month period each before she hopefully finds a solution [23]. The U.S. healthcare system spends approximately $4 billion diagnosing and treating PCOS annually, and patients can spend over a year finding treatment [18]. While gynecologists consult with the patient to provide the best OCP recommendation, the highly variable presentation of PCOS, particularly across ethnic groups, in conjunction with how poorly understood the condition is, can make selecting an effective OCP seem arbitrary. Prescribing the most effective OCP option for PCOS patients is a frustrating and expensive process in need of a more streamlined solution.

## 2.2   Federated Learning Overview

FL is a subfield of privacy-preserving ML which enables multiple clients to train a model collaboratively without exchanging datasets with one another. Training consists of a number of communication rounds. In each communication round, a central server randomly selects a subset of the clients and broadcasts the global model to each of them. Each selected client trains on its local dataset and sends its newly learned model parameters back to the server. Then, the server selects global parameters by aggregating the local parameters in a manner that approximately minimizes overall loss.

We consider four FL approaches: FedAvg [24], FedAvgM [25], FedProx [26], and FedAdam [27]. We designate the first approach as a baseline and select the latter three approaches to handle heterogeneous client datasets. Technical details of these approaches can be found in the Appendix.

## 2.3   Federated Learning Applied to Medicine

FL models optimize dataset diversity, prediction accuracy, and data privacy, making FL the ideal method to harness medical data for improving patient outcomes. The highly distributed nature of medical data limits the application of non-federated ML algorithms, which require storage of data on a central server for training. In practice, this means non-federated models have smaller datasets to train on and lack sufficient training signal. FL models offer improved accuracy by aggregating training results from local datasets into a global model. Importantly, the local data storage units — i.e., client devices — in FL models only exchange summary statistics such as each local model's parameters and dataset size. Privacy is facilitated because local devices can train on their data and transfer training results back to the central server without directly uploading the underlying data. In contrast, non-federated ML approaches house the model and data on a single server, meaning they require access to a massive, central dataset in order to train. These datasets are typically provided by telemedicine companies, such as GoodRx, which are unregulated by HIPAA security rules, and health information companies, which collect proprietary medical data to achieve clinical data interoperability [28], [29]. By sending the model to clients instead of sending the data to the server, FL reduces risk of corruption from a central data controller.

# 3   Experiments

As a proof of concept, we develop FL models which learn OCP treatments based on synthetic patient data. Evaluation of this model is designed to serve as a stepping stone for future FL models learned on real patient data. Using synthetic data is necessary as currently no real patient datasets exist for the task of predicting PCOS treatment.

## 3.1   Generation of Synthetic Data

### 3.1.1   Patient Profiles

For each dataset, we randomly generate patient profiles for 12 clients. These clients represent gynecology practices or hospitals. Each patient profile contains blood test results of seven key metrics

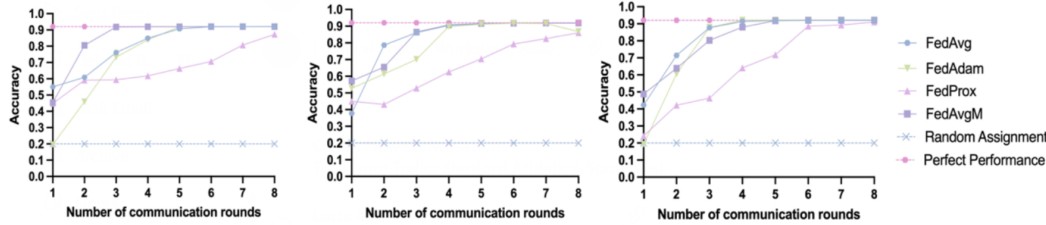

Figure 1: Accuracies observed across FL algorithms. From left to right, these plots depict accuracy on IID data with the same dataset size, non-IID data with the same dataset size, and non-IID data with different dataset sizes. Accuracies for each communication round are obtained by taking the weighted average of accuracies across all clients.

associated with PCOS: luteinizing hormone (LH) to follicle-stimulating hormone (FSH) ratio, total testosterone level, dehydroepiandrosterone (DHEA-S) level, prolactin level, androstenedione level, estradiol level, and anti-müllerian hormone (AMH) level [30]. The profile also includes which OCP was effective in treating the patient. For each metric, we identified the range associated with PCOS diagnosis (see Table 6, column 2) and then split the total range into various sub-ranges to be correlated with one of five OCP options (see Table 7). These sub-ranges were determined arbitrarily as existing literature lacks data on metric distribution. Not every OCP correlates with a specific metric; for example, Apri works best for patients with a LH-FSH ratio within $2 - 2.5$, but lacks correlation with the patient's prolactin or estradiol levels.

### 3.1.2   Generation of Random IID and Non-IID Data

We consider datasets where distribution across each client is IID and datasets where distribution is non-IID. Investigating effects of non-IID data is necessary due to demographic differences across gynecology practices and the varied distribution of PCOS phenotypes [31]. To generate IID data, we assume that clients prescribe each OCP with equal probability. To create synthetic patient profiles, we first select an OCP for each patient independently and uniformly at random. We then sample hormone metrics uniformly at random for each profile using the range associated with the patient's designated OCP (see Table 7). To generate non–IID data, for each client, we first sample a multinomial probability distribution using a Dirichlet distribution with $\alpha = 1$ as the conjugate prior, as in [25]. For each patient, we sample an OCP from this multinomial probability distribution and, as in the IID case, sample hormone metrics uniformly at random using the range associated with the selected OCP. Using a different probability distribution over OCPs for each client allows us to model differences between practices and hospitals.

We inject random noise into our data to mimic confounding variables that impact the optimal OCP treatment. With probability $0.1$, after sampling all hormone levels for a patient, we change their assigned OCP to one sampled uniformly at random from all $5$ available OCP options. This makes our dataset more difficult, capping a model's maximum accuracy at $0.92$. A model achieves this maximum accuracy if it correctly predicts treatment for all unchanged patients and randomly guesses for the $10\%$ of changed patients (with $0.2$ accuracy).

### 3.1.3   Variation in Dataset Size

Clientele and patient numbers vary between gynecology practices. We generate training datasets containing 12,500 patient profiles per client and training datasets of varying sizes per client, where the number of patients is sampled randomly from $Uniform(200, 20000)$. Additionally, we create validation and test datasets for each client that are $25\%$ the size of the test dataset.

Table 1: Accuracies recorded after the last communication round

| Dataset/Algorithm | FedAvg | FedAvgM | FedProx | FedAdam |
|---|---|---|---|---|
| IID data + same dataset size | 0.9211 | 0.9200 | 0.873 | 0.9201 |
| Non-IID data + same dataset size | 0.9181 | 0.9186 | 0.8606 | 0.8676 |
| Non-IID data + dif dataset size | 0.9214 | 0.9209 | 0.9105 | 0.9220 |

## 4 Results

### 4.1 Discussion of Results

We train a model on four different FL algorithms: FedAvg, FedAvgM, FedProx, and FedAdam, which are described in Tables 2, 3, 4, and 5, respectively. First, all four approaches achieve the maximum performance of 92% accuracy on most datasets (see Table 1), meaning the model correctly solves the drug prediction task and randomly guesses on the 10% of noisy patients. Performance is also comparable between IID and non-IID data, despite FL performance traditionally degrading on non-IID data. This shows promise for training on real-world gynecology data, which is likely to consist of non-IID datasets of different sizes. In all four algorithms we test, non-IID data distributions with identical dataset sizes consistently perform slightly worse than the other two dataset types.

Second, only a limited amount of training is needed to achieve the maximum performance of 92% accuracy (see Figure 4). While some algorithms (FedProx) are slower than others (FedAvg) to converge, all tend to converge by the eighth round of training. This demonstrates that the task can be solved with limited computational resources. With approximately 92% accuracy, these basic models predict an OCP that is $4.6$ times more likely to be effective for the patient than randomly selecting a treatment (see Random Assignment in Figure 4). This provides a sufficient proof of concept that FL can significantly improve PCOS patient outcomes, reducing the need for patients to try out multiple drugs and incur the associated costs.

## 5 Conclusions

### 5.1 Limitations

Our results should be interpreted cautiously since we learn on synthetic data. This data may not match that of real gynecology practices, particularly with respect to the number of clients, size of client datasets, and feature distributions. In comparison to our 12 clients and eight communication rounds (with six clients randomly selected per round), typical cross-device FL systems select around $50 - 5,000$ clients per round and take $500 - 10,000$ rounds to converge [32].

The models must begin naively as currently no substantial research exists linking these hormone metrics with OCP performance. When deployed, the model can reveal features which consistently demonstrate low correlations with OCP effectiveness, and this non-correlation can be validated by analysis of medical professionals. The patient profile can be enhanced to include additional factors such as age, BMI, and the presence or absence of PCOS-associated symptoms such as mood disorders, acne, hirsutism, and irregular menstrual cycles. Including patient-reported features into the model improves predictions to better reflect patient disposition and key symptom concerns.

Finally, what the model considers to be the best OCP for the patient — the last option they tried at least six months ago — may be an incorrect assumption for some patient profiles. High healthcare costs can prevent the patient from scheduling a follow-up appointment when their initial OCP is ineffective, or they may forego the optimal OCP because of insurance and pricing constraints.

### 5.2 Further Directions

Challenges that may hinder model performance in realistic settings include partially labeled data and adversarial attacks. Patient data supplied by multiple sites may contain missing or incomplete observations, and deciding how to discard or note absence of an observation during training is a relevant area to explore next [33], [34], [35]. While our selected algorithms address data heterogeneity, they assume the clients and server are trustworthy actors. Algorithms which address privacy, such as

FedPerm [36], can handle a server which attempts to extract clients' data by inspecting model updates sent by participants. A client may also reconstruct another client's private data via gradient inversion attacks [37] or poison the global model by sending manipulated parameter updates. FL is highly compatible with additional privacy features such as homomorphic encryption and secure aggregation, which prevent the server or clients from inferring information about data based on model updates [38], [39], [40]. The security of PCOS patient data in our model can be improved by incorporating these techniques; for example, Salvia is an extension that can securely aggregate updates received from clients and tolerate various percentages of corrupt users [41] [42].

Our work reports only aggregated model performance across all local clients. In a deployment setting, it is important to consider how performance varies among demographic groups. Approaches such as q-FedAvg [43], which we did not consider, handle situations in which the model underperforms on certain clients. Assuming certain hormonal metrics vary by demographic, models must ensure equitable outcomes for each client's patient population. Our aggregation function obtains a global accuracy by weighting each client's accuracy by its dataset size, meaning each patient profile is equally influential to training and the overall classification error is prioritized. Incorporating an algorithm to improve subpopulation accuracy, such as MultiAccuracyBoost [44], offers our model a fairness guarantee [45].

There is also the question of system heterogeneity, which our experiments do not address. Resource-constrained devices may lag due to connectivity issues or variable speeds, and hardware constraints limit the amount of data that can be stored in memory. Given FedAvg's policy of dropping client devices that fail to complete the required number of epochs within a certain timeframe, important data is excluded during training and fair resource distribution across devices is compromised. Further testing should be conducted to ensure the model accurately accommodates devices with varying computational capacities; for example, testing can employ a complete implementation of FedProx, which incorporates partial updates made by "straggler" devices [24].

Women's reproductive health is an area of medicine with massive amounts of untapped data and serious privacy requirements. Even if our model, when deployed on real PCOS patient data, does not detect pertinent correlations between hormone metrics and the effectiveness of OCP treatments, we hope to illuminate FL as an approach to address the many other frontiers in women's health research.

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

# A  Appendix

## A.1  Federated Learning Algorithms

In our work, we explore four FL approaches to robustly evaluate the applications of FL to PCOS patient data. All FL approaches that we consider follow the procedure described in the "FL Overview" section and summarized in Algorithm 1 below. Some notation is adapted from [27].

---

**Algorithm 1** Federated Learning Approaches

---

**Require:** $T$, the number of communication rounds, $N$, the total number of training examples across all clients, and $K$, the number of local SGD epochs.

Initialize $W$, the global model weights
**for** $t = 0, ..., T-1$ **do**
    Sample a set $C$ of clients
    **for** client $i \in C$ **do**
        Broadcast $W$ to client $i$
        Let $\mathcal{L}_i$ be the loss function
        Let $n_i$ be the number of local training examples
        $w_{i,opt} = \texttt{optimize}(\mathcal{L}_i, K)$                          ▷ Run SGD locally for $K$ epochs
        $\Delta_i = w_{i,opt} - W$                               ▷ Delta from global model
        $p_i = n_i/N$
    **end for**
    $\Delta = \sum_{i=1}^{|C|} p_i \Delta_i$                                    ▷ Average the deltas
    Update the global model $W$ using $\Delta$
**end for**

---

Table 2: Steps required for one communication round of FedAvg. Notation defined in Algorithm 1.

| | |
|---|---|
| Loss function on $i$th client | $\mathcal{L}_i = f_i(w_i)$ |
| Optimal weights on $i$th client | $w_{i,opt} = \texttt{optimize}(\mathcal{L}_i, K)$ |
| Delta from the global model on $i$th client | $\Delta_i = w_{i,opt} - W$ |
| Delta averaged across all clients | $\Delta = \sum_{i=1}^{|C|} p_i \Delta_i$ |
| Global model update | $W \leftarrow W + \Delta$ |

Table 3: Steps required for one communication round of FedAvgM. Notation defined in Algorithm 1. Steps that differ from those of FedAvg are bolded.

| | |
|---|---|
| Loss function on $i$th client | $\mathcal{L}_i = f_i(w_i)$ |
| Optimal weights on $i$th client | $w_{i,opt} = \texttt{optimize}(\mathcal{L}_i, K)$ |
| Delta from the global model on $i$th client | $\Delta_i = w_{i,opt} - W$ |
| Delta averaged across all clients | $\Delta = \sum_{i=1}^{|C|} p_i \Delta_i$ |
| **Global model update** | $v \leftarrow \beta v + \Delta$ 
 $W \leftarrow W + v$ 
 where $\beta$ is a hyperparameter |

### A.1.1   FedAvg (see Table 2)

FedAvg (Federated Averaging) is the simplest FL approach, in which a single global model is broadcast from a server to a randomly selected subset of clients in each communication round [24]. Each client performs a few epochs of stochastic gradient descent (SGD) using its own local data on its own copy of the global model. Finally, the updated global models are broadcast back to the server, which updates the global model as an average of all client models, weighted by the number of training examples on each client.

We use FedAvg as our baseline approach for minimizing loss across clients. While FedAvg demonstrates strong performance on the average distribution of all clients, the technique does not explicitly address cases of non-IID data between clients. This approach raises concern because patient data distributions likely differ across clients, particularly due to patient demographics. We therefore experiment with three additional approaches designed to address data heterogeneity. Note that in our setting, $f_i(w_i)$ refers to a cross entropy loss.

### A.1.2   FedAvgM (see Table 3)

Inspired by the success of applying momentum updates to SGD [46], FedAvgM (Federated Averaging with Server Momentum) updates the global model by accumulating the gradients broadcast from the clients with a momentum term (Equation 3) [25]. FedAvgM is applicable to our setting as the momentum term potentially makes global updates less susceptible to noisy variations in local model updates caused by non-IID data.

Table 4: Steps required for one communication round of FedProx. Notation defined in Algorithm 1. Steps that differ from those of FedAvg are bolded.

| | |
|---|---|
| **Loss function on $i$th client** | $\mathcal{L}_i = f_i(w_i) + \frac{\mu}{2}\|W - w_i\|^2$ 
 where $\mu$ is a hyperparameter |
| Optimal weights on $i$th client | $w_{i,opt} = \texttt{optimize}(\mathcal{L}_i, K)$ |
| Delta from the global model on $i$th client | $\Delta_i = w_{i,opt} - W$ |
| Delta averaged across all clients | $\Delta = \sum_{i=1}^{|C|} p_i \Delta_i$ |
| Global model update | $W \leftarrow W + \Delta$ |

Table 5: Steps required for one communication round of FedAdam. Notation defined in Algorithm 1. Steps that differ from those of FedAvg are bolded.

| | |
|---|---|
| Loss function on $i$th client | $\mathcal{L}_i = f_i(w_i)$ |
| Optimal weights on $i$th client | $w_{i,opt} = \texttt{optimize}(\mathcal{L}_i, K)$ |
| Delta from the global model on $i$th client | $\Delta_i = w_{i,opt} - W$ |
| Delta averaged across all clients | $\Delta = \sum_{i=1}^{|C|} p_i \Delta_i$ |
| **Global model update** | $m \leftarrow \beta_1 m + (1 - \beta_1)\Delta$ 
 $v \leftarrow \beta_2 v + (1 - \beta_2)\Delta^2$ 
 $W \leftarrow W + \eta \frac{m}{\sqrt{v} + \tau}$ 
 where $\beta_1$, $\beta_2$, $\eta$, and $\tau$ are hyperparameters |

### A.1.3   FedProx (see Table 4)

FedProx improves on FedAvg's tendency to favor certain devices' performance, introducing a proximal term in the loss that penalizes large changes to weights in the global model [26]. This prevents any single client from deviating from the global model too much, making global updates less susceptible to noisy variations between non-IID clients. The original FedProx algorithm also includes a mechanism for incorporating partial updates from client device failures due to system heterogeneity, but for our purposes, we only consider the proximal term which handles data heterogeneity.

### A.1.4   FedAdam (see Table 5)

FedAdam belongs to a family of FL algorithms, termed FedOpt, which improve on FedAvg [27]. FedAdam uses global updates inspired by the Adam optimizer [47] as opposed to SGD updates, and it applies to our use case because adaptive optimization methods respond well to mild heterogeneity between clients.

## A.2   Synthetic Data Correlations

Tables 6 and 7 contain the correlation ranges that we used to generate our synthetic data.

## A.3   Experimental Details

We use the Flower framework [41], a scalable FL framework that handles heterogeneous data sources and offers several built-in FL algorithms. We implement a simple fully-connected neural network as our backbone architecture. The architecture consists of two hidden layers, each of which contains between 10 and 20 neurons. We apply a ReLU nonlinearity between each layer, and after the final layer, we apply a softmax function to normalize the predicted class scores. We train the model for eight communication rounds in total. In the first communication round, two randomly-selected clients participate, and in later communication rounds, half of the clients are randomly selected to participate. Participating clients train for three epochs on their local datasets per round. Each client's copy of the neural network trains using a cross entropy loss and SGD with momentum with the following hyperparameters: learning rate of $0.0008$, momentum of $0.87$, and batch size of $32$.

The FedProx algorithm adds a proximal term to the loss when training each client. We set the coefficient on this proximal term to $0.3$. Additionally, we set $\beta = 0.8$ in FedAvgM. In FedAdam, we set $\beta_1 = 0.9$, $\beta_2 = 0.99$, $\eta = 0.1$, and $\tau = 1e^{-9}$.

Table 6: Chart of synthetic correlation for each blood panel. Each blood panel metric from the patient profile is mapped to the typical range of that metric in a PCOS patient, followed by the synthetic correlation of the metric with its most effective OCP option.

| Patient Profile Metric | Range of PCOS Diagnosis | Implanted Correlations |
|---|---|---|
| LH-FSH Ratio | 2-3.5 [48] | Apri: 2-2.5
Yaz: 2.6-3
Cyclen: 3.1-3.5 |
| Total Testosterone | 86-150 ng/dl [49] | Cyclen: 86-100.9 ng/dl
Tri-cyclen: 101-110.9 ng/dl
Diane-35: 111-120.9 ng/dl
Apri: 121-130.9 ng/dl
Yaz: 131-150 ng/dl |
| DHEA-S | 200-430 ug/dl [50] | Apri: 200-300.9 ug/dl
Cyclen: 301-350.9 ug/dl
Tri-cyclen: 351-400.9 ug/dl
Diane-35: 401-430 ug/dl |
| Prolactin | 25-40 ng/ml [51] | Diane-35: 25-30.9 ng/ml
Cyclen: 31-35.9 ng/ml
Yaz: 36-40 ng/ml |
| Androstenedione | 0.4-2.7 ng/ml [49] | Tri-cyclen: 0.4-0.7 ng/ml
Yaz: 0.8-1.0 ng/ml
Apri: 1.1-1.5 ng/ml
Cyclen: 1.6-2.0 ng/ml
Diane-35: 2.1-2.7 ng/ml |
| Estradiol | 60-120 pg/ml [52] | Yaz: 60-80.9 pg/ml
Cyclen: 81-100.9 pg/ml
Tri-cyclen: 101-120 pg/ml |
| Anti-Mullerian Hormone (AMH) | 5-10 mcg/L [53] | Yaz: 5-6.5 mcg/L
Diane-35: 6.6-8 mcg/L
Apri: 8.1-10 mcg/L |

We report results of our approaches on multiple datasets to evaluate the effects of both IID and non-IID data distributions and differing dataset sizes. These datasets include:

1. IID data among all 12 clients where each client has $12,500$ (train) patients and a quarter as many test patients.

2. Non-IID data where each client has $12,500$ (train) patients and a quarter as many test patients.

3. Non-IID data where each client has between $200$ and $20,000$ (train) patients and a quarter as many test patients.

We normalized all hormone levels to a range between $-0.5$ and $0.5$ to facilitate stable training.

Table 7: Chart of synthetic correlation for each OCP option. Each OCP option is mapped to the range of blood panel metrics used for synthetic data generation. [NC] denotes no synthetic correlation exists in this dataset between the OCP and hormonal metric.

| OCP | Model Correlation Ranges |
| --- | --- |
| Apri | **LH-FSH ratio:** 2-2.5
**Total testosterone:** 121-130.9 ng/dl
**DHEA-S:** 200-300.9 ug/dl
**Prolactin:** [NC] 25-40 ng/ml
**Androstenedione:** 1.1-1.5 ng/ml
**Estradiol:** [NC] 60-120 pg/ml
**Anti-Mullerian (AMH):** 8.1-10 mcg/L |
| Cyclen | **LH-FSH ratio:** 3.1-3.5
**Total testosterone:** 86-100.9 ng/dl
**DHEA-S:** 301-350.9 ug/dl
**Prolactin:** 31-35.9 ng/ml
**Androstenedione:** 1.6-2.0 ng/ml
**Estradiol:** 81-100.9 pg/ml
**Anti-Mullerian (AMH):** [NC] 5-10 mcg/L |
| Tri-cyclen | **LH-FSH ratio:** [NC] 2-3.5
**Total testosterone:** 101-110.9 ng/dl
**DHEA-S:** 351-400.9 ug/dl
**Prolactin:** [NC] 25-40 ng/ml
**Androstenedione:** 0.4-0.7 ng/ml
**Estradiol:** 101-120 pg/ml
**Anti-Mullerian (AMH):** [NC] 5-10 mcg/L |
| Yaz | **LH-FSH ratio:** 2.6-3
**Total testosterone:** 131-150 ng/dl
**DHEA-S:** [NC] 200-430 ug/dl
**Prolactin:** 36-40 ng/ml
**Androstenedione:** 0.8-1.0 ng/ml
**Estradiol:** 60-80.9 pg/ml
**Anti-Mullerian (AMH):** 5-6.5 mcg/L |
| Diane-35 | **LH-FSH ratio:** [NC] 2-3.5
**Total testosterone:** 111-120.9 ng/dl
**DHEA-S:** 401-430 ug/dl
**Prolactin:** 25-30.9 ng/ml
**Androstenedione:** 2.1-2.7 ng/ml
**Estradiol:** [NC] 60-120 pg/ml
**Anti-Mullerian (AMH): 6.6-8 mcg/L** |

