# OpenReview forum: "Federated Learning on Patient Data for Privacy-Protecting Polycystic Ovary Syndrome Treatment"
_NeurIPS.cc/2022/Workshop/SyntheticData4ML — Neurips 2022 SyntheticData4ML_

### Official Review · Reviewer_ZSTT · 2022-10-10
**A new application of FL in healthcare**

**Rating:** 6
**Confidence:** 4

**Review:**

This work studies a new application of federated learning in healthcare, or more specifically, women's endocrinology. The task is to predict the optimal drug with PCOS, and FL can address the privacy concerns. It conducts experiments of several FL algorithms on health datasets that the authors created, and proper results are obtained. Although the idea is straightforward, it shows significant improvement in PCOS patient outcomes. Finally, the paper discusses limitations and future directions.

---

### Official Review · Reviewer_9KZU · 2022-10-18
**Interesting study on the comparison of different federated learning methodologies in the context of PCOS data**

**Rating:** 8
**Confidence:** 5

**Review:**

The paper highlights an important topic of PCOS data and lack of data and investigation in this area. The authors generate synthetic data using different methodologies (IID and non-IID) and aim to evaluate and compare the differences between the different federated learning methodologies. The paper presents an interesting dataset and comparison between the federated learning methodologies in the context of a unique dataset.

The only limitation of this study I can think of is the construction of the synthetic dataset as well as the high AUC (92%) which I believe is a direct result of how the synthetic dataset was constructed. This level of performance may not be realistic to expect on real datasets and may impact the comparison and inferences made about the federated learning methodologies. It would be intriguing to see the same experiments applied to synthetic datasets that mimic synthetic/real datasets that are more "realistic" or at least some acknowledgment from the authors about this limitation. Alternatively, if real data on PCOS is available on a limited scale, it would be great to see use of open-source synthesizers (Gretel, MIT SDV, etc) to generate more data for augmentation on PCOS for example depending on if this is the case.

---

### Official Review · Reviewer_q6eg · 2022-10-18
**Interesting application, but paper making some strange claims**

**Rating:** 6
**Confidence:** 4

**Review:**

The paper presents an interesting and well-motivated example of use of synthetic data for developing federated learning with medical data. Methodologically, there is not much novelty. The paper makes a few questionable claims that should be corrected before publication. The writing is OK, although personally I would have preferred to read more about the data synthesis process and less about the medical details of the application.

Specifically, the questionable claims that should be corrected are:

> By randomly selecting which clients participate in each learning round, FL incorporates elements of RCTs for greater statistical power.

I find the comparison of client subsampling to RCT far-fetched: client subsampling is still observational and there is no intervention. There is no control either. The claim that this leads to greater statistical power would need strong supporting evidence. I would recommend deleting the whole claim.

> FL models optimize [...] data privacy. [...] Privacy is maintained because local devices can train on their data and transfer training results back to the central server without exposing the underlying data.

Vanilla FL is not a foolproof privacy technology, as it is susceptible to gradient inversion attack to recover client data. If you wish to guarantee privacy, you need to combine FL with something like differential privacy. I would strongly recommend adding a mention of this to avoid overselling your approach.

---

### Meta-Review · Area_Chair_YMQF · 2022-10-18

**Recommendation:** Accept

**Review:**

The AC agrees with Reviewer q6eg on the questionable claims related to imitating RCT and privacy preservation. Please make revision in the Camera-ready version.